# Association of Trimethylamine N-Oxide with Normal Aging and Neurocognitive Disorders: A Narrative Review

**DOI:** 10.3390/brainsci12091203

**Published:** 2022-09-06

**Authors:** Xiangliang Chen, Mengmeng Gu, Ye Hong, Rui Duan, Junshan Zhou

**Affiliations:** Department of Neurology, Nanjing First Hospital, Nanjing Medical University, Changle Road 68, Nanjing 210006, China

**Keywords:** neurocognitive disorders, brain aging, cognitive dysfunction, trimethylamine-N-oxide, inflammation

## Abstract

Aging-related neurocognitive disorder (NCD) is a growing health concern. Trimethylamine-N-oxide (TMAO), a gut microbiota-derived metabolite from dietary precursors, might emerge as a promising biomarker of cognitive dysfunction within the context of brain aging and NCD. TMAO may increase among older adults, Alzheimer’s disease patients, and individuals with cognitive sequelae of stroke. Higher circulating TMAO would make them more vulnerable to age- and NCD-related cognitive decline, via mechanisms such as promoting neuroinflammation and oxidative stress, and reducing synaptic plasticity and function. However, these observations are contrary to the cognitive benefit reported for TMAO through its positive effects on blood–brain barrier integrity, as well as from the supplementation of TMAO precursors. Hence, current disputable evidence does not allow definite conclusions as to whether TMAO could serve as a critical target for cognitive health. This article provides a comprehensive overview of TMAO documented thus far on cognitive change due to aging and NCD.

## 1. Introduction

The increased pace of aging has been both a challenge and a triumph for public health. Globally, 1 in 6 people will be aged 60 years or above by 2030 [1]. Advanced age is one of the strongest risk factors for neurocognitive disorders (NCD) such as Alzheimer’s disease (AD) and poststroke cognitive impairment (PSCI) [2]. Driven by the aging population, the number of individuals with dementia is doubling every 20 years, and even among dementia-free seniors, 1 in 5 have mild NCD syndromes [3]. Patients with NCD typically exhibit longitudinal declines in cognitive and functional abilities, posing an increased risk for higher health expenditure [4] and all-cause mortality [2]. Although age increases risk, there are marked individual differences in the vulnerability for older adults in terms of NCD. Indeed, the onset of these disorders could be prevented or delayed, given that a wide range of predictive features, such as bio-behavioral factors, psychosocial characteristics, and cardiovascular diseases, are potentially modifiable [5,6]. Among them, gut microbiota and their metabolic products, including trimethylamine-N-oxide (TMAO), have recently emerged as a promising disease modifier and might play a putative role in aging and the development and progression of NCD [7,8,9].

TMAO is synthesized from the oxidation of trimethylamine (TMA) by hepatic flavin monooxygenases (FMO), and TMA is a microbiota-derived metabolite, generated in the gut from dietary precursors, mainly choline, L-carnitine, and betaine [10]. The TMAO level is affected by dietary intake, renal clearance, and host and microbial enzymes [11,12]. In contrast with a positive correlation observed between the dietary intake of TMAO precursors and cognitive function [13,14,15], elevated TMAO levels are detected in healthy aging [16] and associated with the pathogenesis of several diseases—for example, metabolic abnormalities, autoimmune disorders, colon cancer, and most prominently, atherosclerotic diseases [17,18]. A circulating TMAO level in the range of 1.5 µM to 10.5 µM has been shown to have a dose-dependent relationship with incident cardiovascular risk [19], the exposure of which serves as a vital target for cognitive health [20]. On top of that, TMAO itself contributes to cognitive decline both with aging [21] and NCD [22], suggesting an effect of higher TMAO levels on NCD per se, as well as their risk factors. Evidence shows that TMAO may be involved in the processes of brain aging and cognitive impairment via promoting neuroinflammation and oxidative stress, as well as reducing synaptic plasticity and function [23]. Despite this, opposing findings that TMAO had positive effects upon the blood–brain barrier (BBB) integrity and murine cognitive function in response to inflammatory challenge [24] illuminated the complex nature of the relationship that remains to be elucidated.

To date, whether TMAO could play a critical role in the pathophysiology of cognitive impairment due to aging and NCD, thus affecting disease development, or whether it could be a mediator contributing to disease progression, or is just a consequential marker, is still unclear. Therefore, the present review aims to summarize current evidence regarding the potential relationship of TMAO with normal cognitive aging and NCD.

## 2. Normal Cognitive Aging

The TMAO level may rise during the normal aging process [21,25] (Table 1). Healthy individuals aged above 65 years had significantly increased plasma TMAO levels, with a mean plasma concentration of 9.8 µM, in comparison to 4.4 µM in adults aged 45 to 64 years, and 2.8 µM in adults aged 18 to 44 years, revealing that TMAO was positively related to age (*r*^2^ = 0.161, *p* < 0.001) [25]. Moreover, brain TMAO levels were also higher in old vs. young mice, and were highly correlated to the levels in circulation [21], indicating a direct effect of TMAO on the brain and cognitive function.

Indeed, in healthy middle-aged to older adults, circulating TMAO levels inversely predicted working memory and fluid cognition independent of traditional risk factors [21]. In mice, 16-week treatment of TMAO at a concentration of 1.5% could induce and aggravate brain aging and aging-related cognitive dysfunction as a result of neuron senescence, and the underlying mechanism would be the mitochondrial impairments driven by oxidative stress and the reduced expression of synaptic plasticity-related proteins by inhibiting the mammalian target of rapamycin (mTOR) signaling pathway [25]. Likewise, preexisting higher circulating TMAO may sensitize sevoflurane-induced cognitive impairment in aged rats, probably via downregulating antioxidant enzyme methionine sulfoxide reductase A in the hippocampus, then leading to microglia-mediated neuroinflammation [26]. Furthermore, TMAO would induce aging-like cognitive impairments in young animals, as TMAO-supplemented young adult mice performed worse on the novel object recognition test compared to the controls, with higher concentrations of pro-inflammatory cytokines and the reactive astrocyte marker, suggesting that TMAO might mediate cognitive aging by inducing neuroinflammation and astrocyte activation [21] (Table 1).

**Table 1 brainsci-12-01203-t001:** Main studies of TMAO involved in healthy aging.

Authors (Year)	Subjects/Models	Interventions	Main Related Findings	Conclusions
Li et al. (2018) [25]	The elderly > 65 years (*n* = 141), middle-aged adults between 45–64 years (*n* = 118), and young adults aged 18–44 years (*n* = 168)	N/A	Plasma TMAO: higher in the elderly than in the middle-aged and young groups (9.83 ± 10.63 vs. 4.42 ± 4.39 vs. 2.85 ± 3.10 µM), positively related to age (*r*^2^ = 0.1610, *p* < 0.001) TMAO precursors (choline, carnitine, betaine, and butyrobetaine): did not differ significantly in the three groups	The plasma level of TMAO increased with age in humans, but TMAO precursors did not increase with age
	24-week-old male SAMP8 (*n* = 12) and SAMR1 (*n* = 12) mice	1.5% TMAO in drinking water vs. sterile water for 16 weeks	Plasma TMAO: increased with age in SAMR1 and SAMP8 mice Cognition (Y-maze, novel object recognition, and Morris water maze): aggravated by TMAO Hippocampus: TMAO increased senescent neurons in CA3 region; damaged ultrastructure of chemical synapses in CA1 region; increased oxidative stress; reduced expressions of synaptic plasticity-related proteins; down-regulated activity of the mTOR signaling pathway	TMAO could deteriorate brain aging and cognitive function by promoting neuron senescence, damaging synapses, down-regulating the expression of synaptic plasticity-related proteins, and inhibiting the mTOR signaling pathway
Li et al. (2019) [27]	20 ± 2 g male ICR mice (*n* = 10), aging induced by daily intraperitoneally injected D-gal (120 mg/kg) and NaNO_2_ (90 mg/kg) for 3 months	On the 25th day of aging induction, daily intragastric administration of vitamin E (100 mg/kg) or Fructus Ligustri Lucidi aqueous extract (4.9 g/kg) for 65 days	Serum TMAO: aging mice were higher than controls (0.38 ± 0.08 vs. 0.24 ± 0.07 μM); could be decreased to 0.28 ± 0.09 μM after Fructus Ligustri Lucidi administration; associated with several bacterial taxa (Sutterella↑, Unclassified_Clostridiales↑, Corpococcus↑, Clostridium↑, Unclassified_S24-7↑, SMB53↑, Aldercreutzia↑, Oscillospira↑, Desulfovibrio↑, Bifidobacterium↓, and Lactobacillus↓)	Fructus Ligustri Lucidi may have an anti-aging effect by regulating the imbalance in the intestinal microbiota and the increase in serum TMAO levels in aging mice induced by D-gal and NaNO_2_
Brunt et al. (2021) [21]	Middle-aged and older healthy adults aged 50–79 years (*n* = 103), and young healthy adults aged 18–27 years (*n* = 22)	N/A	Plasma TMAO: higher in the middle-aged and older adults vs. young adults; not differ between men (7.4 ± 7.4 μM) and women (6.2 ± 6.6 μM) TMAO precursors: higher plasma choline in the middle-aged and older adults vs. young adults (13.9 ± 6.1 vs. 7.9 ± 3.2 μM), comparable L-carnitine and betaine Cognition (NIH toolbox cognition battery test, trail-making test): inversely related to TMAO in subdomains of working memory, episodic memory, and fluid cognition	Concentrations of TMAO increased with aging and had no sex differences Plasma TMAO could predict working memory and fluid cognition independent of cardiovascular risk in middle-aged to older adults
	Male C57Bl/6 young mice at 8 weeks of age (*n* = 34) and old mice at 20–24 months of age (*n* = 16)	a defined 0.07% choline diet with or without 0.12% TMAO for 6 months	Plasma and brain TMAO: highly correlated, higher in old vs. young mice, greater in TMAO-supplemented mice vs. controls Plasma and brain TMAO precursors (choline, betaine, and L-carnitine): not correlated, differed in old vs. young mice, and not affected by TMAO supplementation Cognition (novel object recognition): TMAO impaired memory and spatial learning Whole-brain lysates: TMAO-supplemented mice had increased IL-1β, TNF-α, phosphorylated NF-κB, and reactive astrocyte marker LCN2 vs. controls Cultured human astrocytes: increased LCN2 and CD44 if treated with 100 μM TMAO	TMAO may cross the BBB to a greater extent than TMAO precursors Increased plasma and brain levels of TMAO, induced by either natural aging or supplementation, could cause cognitive decline accompanied by astrocyte activation and neuroinflammation

Abbreviations: ↑, increase; ↓, decrease; CA, cornu ammonis; CD44, cluster of differentiation 44; LCN2, lipocalin 2; mTOR, the mammalian target of rapamycin; NF-κB, nuclear factor-κB; NIH, National Institutes of Health; SAMP8, senescence-accelerated prone mouse strain 8; SAMR1, senescence-accelerated mouse resistant; TMAO, trimethylamine-N-oxide; TNF-α, tumor necrosis factor α.

Therefore, the potential of TMAO as a prevention and/or treatment target for cognitive declines in aging has been exploited (Table 1). The aging mice demonstrating deficits in memory and cognitive function revealed an improvement in cognition after *Fructus Ligustri Lucidi* (i.e., the ripe fruit of *Ligustrum lucidum Ait*) treatment, possibly by lowering oxidative stress subsequent to decreased circulating TMAO levels via the altered gut microbiota, characterized as a reduction in *Bifidobacterium* and *Lactobacillus*, and an increase of the *Sutterella*, *Unclassified_Clostridiales*, *Corpococcus*, and *Clostridium*, among others [27]. However, nutritional intake of TMA precursors might have cognitive protection capacities. For instance, a randomized clinical trial has shown that the supplementation of 2 g L-carnitine taken orally once a day for six months significantly improved the cognitive function in subjects above 100 years of age, showing significant improvements in the mini-mental state examination (MMSE) score (4.10 compared with 0.60) when compared to the placebo group [28].

## 3. Alzheimer’s Disease

In 2017, Del Rio et al. provided the first evidence that TMAO can be detected in the human cerebral spinal fluid (CSF) [29] (Table 2). The presence of TMAO in the CSF might be hepatic-derived, as TMAO could cross the BBB [30], and high amounts of circulating TMAO might be linked to BBB disruption in vivo [31], yet de novo synthesis of TMAO could also be possible since FMO3 has been detected in the adult brain [32]. The CSF levels of TMAO in AD patients were elevated compared to cognitively unimpaired individuals [33] but were similar to that in patients with mild cognitive impairment (MCI) [33], non-AD-related dementia, or other neurological disorders [29]. Consistently, urine TMAO levels were comparable between MCI and AD patients, and both were higher than cognitively healthy controls (mean TMAO levels in the urine, cognitively healthy controls: 10.2 µM, MCI: 19.9 µM, AD: 18.9 µM) [3].

The intriguing role of TMAO in AD etiology was indicated in a 2016 study using a network-based ranking algorithm [22]. Among the known AD-associated microbial metabolites in humans, TMAO ranked as the top one that shared significant genetic commonality with AD. There were nine co-regulated genetic pathways, including pathways related to “AD”, “axon guidance”, “immune systems”, “neuron signaling”, and “lipid and protein metabolism” [22]. Interestingly, in vitro studies demonstrated that TMAO affected amyloid-β (Aβ) conformation and facilitated Aβ aggregation [34,35]; moreover, the reduction of plasma TMAO levels, either by treatment of 3,3-Dimethyl-1-butanol (DMB) or a combination of *Lactobacillus plantarum* and memantine, led to significantly decreased concentration of Aβ42, Aβ40, and Aβ deposition in the hippocampus of APP/PS1 mice [36,37]. However, CSF TMAO levels were not significantly correlated with Aβ42/Aβ40, but with CSF phosphorylated tau (p-tau) and p-tau/Aβ42 [33], the aggregation of which shown to be enhanced by TMAO by using in vitro models [38,39] (Table 2).

An association of TMAO with neurodegeneration has also been displayed alongside the pathological profile, that the CSF TMAO levels were more closely related to the biomarker of axonal injury (CSF total tau and neurofilament light chain protein) than the biomarker of dendritic degeneration (CSF neurogranin) [33]. Meanwhile, circulating TMAO levels have been related to hippocampal neuroinflammation. Downregulation of TMAO levels in plasma could alleviate the neuroinflammatory state of AD model mice, as indicated by the significant reductions in plasma clusterin, together with IL-2, IL-17, and TNF-α levels in the hippocampus [36,37]. Furthermore, TMAO impaired synaptic plasticity in the form of reduced long-term potentiation (LTP) through the endoplasmic reticulum (ER) stress-mediated PERK signaling pathway [40] (Table 2).

The increased plasma TMAO in high-fat feeding dementia-prone (3xtg) mice [41] and choline-supplemented mice [37] also correlated with cognitive impairment. Lowering TMAO by DMB treatment or *Lactobacillus plantarum* supplementation protected hippocampal neuronal integrity and plasticity [37] and ameliorated LTP and cognitive decline in AD transgenic mice [36,37]. Nonetheless, these associations do not reflect causality. According to a recent bidirectional Mendelian randomization study, TMAO and its precursors, including choline, carnitine, and betaine, did not have a causal effect on the risk of AD [42] (Table 2).

**Table 2 brainsci-12-01203-t002:** Main findings of TMAO in relation to AD.

Authors (Year)	Subjects/Models	Interventions	Main Related Findings	Conclusions
Del Rio et al. (2017) [29]	AD (*n* = 22), non-AD related dementia including FTD, CBD, and DP (*n* = 16), age- and sex-matched subjects with other neurological disorders unrelated to demyelinating inflammatory disorders, stroke and neurodegenerative and infective diseases (*n* = 20)	N/A	CSF TMAO: no differences among the three groups (medium CSF TMAO level: AD, 0.52 μM; non-AD dementia, 0.71 μM; others, 0.57 μM)	TMAO could be detected in human CSF, but its levels might be unrelated to different neurological disorders
Sanguinetti et al. (2018) [41]	Male B6129SF2/J mice (*n* = 18 total) and triple transgenic (3xtg) male mice (*n* = 15 total)	High-fat diet-fed vs. normal diet-fed administered from 2 to 8 months of age	Serum TMAO: an elevated tendency in 3xtg mice vs. controls, further elevated in high-fat diet-fed 3xtg mice Cognition (Y-maze): a non-significant 20–40% cognitive decline in 3xtg models	High-fat diet and genetic predisposition led to microbiome-metabolome changes that preceded dementia
Vogt et al. (2018) [33]	AD patients (*n* = 40), MCI patients (*n* = 35), and cognitively-unimpaired controls (*n* = 335)	N/A	CSF TMAO: higher in AD dementia and MCI vs. controls after controlling for age, sex, and *APOE* ε4 genotype; positively associated with age; did not differ between MCI and AD CSF biomarkers: CSF TMAO positively related to p-tau, p-tau/Aβ_42_, t-tau, and NFL, not related to Aβ42/Aβ40 nor neurogranin, associations did not change when including peripheral cardiovascular disease risk factors as covariates	CSF TMAO would be higher in MCI and AD dementia compared to cognitively unimpaired individuals; elevated CSF TMAO might be associated with CSF biomarkers of tau pathology and axonal degeneration
Gao et al. (2019) [36]	3-, 6-, 9-, and 12-month-old male WT and APP/PS1 mice (*n* = 10 each)	With or without 1.0% DMB in drinking water for 8 weeks	Plasma TMAO: WT mice had higher TMAO at 12 vs. 3–9 months, APP/PS1 mice had higher TMAO at 9 vs. 3–6 months and further increased at 12 months, TMAO differences between WT and APP/PS1 mice greatest at 12 months, could be significantly decreased by DMB treatment Cognition (novel object recognition and Morris water maze): inversely correlated with TMAO levels in object recognition memory, spatial learning and memory, and active avoidance in the 9- and 12-month-old mice; DMB ameliorated circulating TMAO levels and cognition deficiencies in APP/PS1 mice Hippocampus: Aβ_1–42_, β-secretase, βCTF levels, IL-2, IL-17, and TNF-α decreased in DMB-treated vs. non-treated APP/PS1 mice	Circulating TMAO levels would increase with age, and be associated with AD-like behavioral and pathological profile of APP/PS1 mice. TMAO reduction by DMB could reverse the upregulation of clusterin levels in the plasma, Aβ_1–42_, β-secretase, βCTF, and proinflammatory cytokines in the hippocampus, and cognition deficiencies in AD model mice
Govindarajulu et al. (2020) [40]	8- and 18-month-old female 3xTg-AD mice and control C57BL/6 mice, 8-month-old female diabetic db/db mice (*n* = 3 each)	An ex-vivo model by incubating wild-type hippocampal brain slices with 50 mM of TMAO for 4–6 h or 0.03% DMSO vehicle (control)	Serum and brain TMAO: higher in 3xTg-AD and db/db mice at 8 months vs. controls, further increased in 3xTg-AD mice at 18 months vs. controls Hippocampal slices: reduced LTP and impaired synaptic transmission through induction of the PERK-EIF2α-endoplasmic reticulum stress signaling axis in TMAO incubated slices vs. controls	TMAO increased in 3xTg-AD and diabetic db/db mice compared to wild-type mice TMAO may induce deficits in synaptic plasticity by the endoplasmic reticulum stress-mediated PERK signaling pathway
Wang et al. (2020) [37]	I. 8-week-old male C57BL/6J mice (*n* = 60 total) II. 6-month-old male WT and APP/PS1 mice (*n* = 15 each)	I. A chow diet supplemented with or without 1% choline for 3 months II. Sterilized PBS, 1 mg/mL memantine, 1 × 10^9^ CFU/mL *L. plantarum*, or a combination of memantine and *L. plantarum* once a day for 12 weeks by oral gavage	Plasma TMAO: higher in choline-treated mice vs. controls, decreased by treatment with *L. plantarum* or a combination of memantine and *L. plantarum* Cognition (novel object recognition and Morris water maze): cognitive declines in choline-treated C57BL/6J mice associated with increased plasma TMAO levels and *L. plantarum* Hippocampus: Aβ plaques, Aβ_1–42_ and Aβ_1–40_ levels, neuron integrity, and plasticity in APP/PS1 mice correlated with circulating TMAO	*L. plantarum* decreased TMAO levels by suppressing gut microbial TMA secretion via remodeling of gut microbiota in transgenic AD mice, thereby attenuating cognitive impairments and pathological deterioration
Yilmaz et al. (2020) [43]	AD patients (*n* = 20), MCI patients (*n* = 10), and healthy controls (*n* = 29)	N/A	Urinary TMAO: higher in AD and MCI than in the controls (18.864 ± 11.571 vs. 19.907 ± 10.822 vs. 10.229 ± 7.735 µM), not differ between AD and MCI	Urine metabolomics may be useful for distinguishing MCI and AD from cognitively healthy controls
Zhuang et al. (2021) [42]	Disease AD data (*n* = 455,258) Human metabolone data (*n* = 2076)	A bidirectional mendelian randomization analysis	Genetically predicted higher TMAO, betaine, carnitine, and choline were not significantly associated with the risk of AD after Bonferroni correction. In the other direction, AD was also not causally associated with levels of TMAO, betaine, carnitine, or choline.	TMAO or its predecessors do not play causal roles in the development of AD

Abbreviations: Aβ, amyloid beta; AD, Alzheimer’s disease; *APOE*, Apolipoprotein E; βCTF, β-secretase-cleaved C-terminal fragment; CBD, corticobasal degeneration; CFU, colony-forming units; CSF, cerebrospinal fluid; DMB, 3,3- dimethyl-1-butanol; DMSO, dimethyl sulfoxide; DP, degenerative parkinsonism; EIF2α, eukaryotic initiation factor 2α; FTD, frontotemporal dementia; IL-2, interleukin-2; IL-17, interleukin-17; MCI, mild cognitive impairment; NFL, neurofilament light; PBS, phosphate buffered saline; PERK, PKR-like endoplasmic reticulum kinase; TMAO, trimethylamine-N-oxide; TNF-α, tumor necrosis factor α; WT, wild type.

## 4. Poststroke Cognitive Impairment

The predictive role of elevated TMAO has been evaluated in the development of PSCI. For patients with first-ever ischemic stroke admitted <7 days of symptom onset, higher plasma TMAO within 24 h of admission may increase the likelihood of PSCI assessed by the MMSE score ≤ 26 at 1 year after stroke (highest TMAO quartile > 7.4 μM vs. lowest TMAO quartile < 3.9 μM: adjusted odds ratio, aOR, 3.304; 95% confidence intervals, 95% CI, 1.335–8.178; *p* = 0.010) [44]. The admission plasma TMAO levels were also higher in patients suffering from a minor stroke in the past 2 weeks who had PSCI defined as < 22 points on the Montreal Cognition Assessment (MoCA) at 6–12 months after stroke onset than those without PSCI (median plasma TMAO: 4.6 μM vs. 3.2 μM; *p* ≤ 0.001), but there were no significant differences in circulating levels of TMAO precursors such as L-carnitine and choline [45] (Table 3).

Nevertheless, clinical findings investigating the relationship between PSCI and TMAO have been inconsistent. Zhong et al. showed that the baseline plasma TMAO level within 72 h of stroke onset was only associated with MMSE-defined PSCI (i.e., MMSE score < 27), with an aOR of 1.33 (95% CI, 1.04–1.72) for each 1-SD increment of TMAO, but there was no significant relationship when PSCI was defined according to MoCA score < 25, while notably, an inverse dose-response relationship was observed for TMAO precursors (choline and betaine) with either MMSE-defined or MoCA-defined PSCI [46]. A possible explanation for this discrepancy, besides the different definitions of PSCI, arises from variations in the timing of blood collection as TMAO levels were observed to elevate within 24 h after symptom onset, but then decreased in the following week, although it increased again after three months [47,48] (Table 3).

Experimental evidence is scarce. In mice undergoing repeated cerebral ischemia-reperfusion injury, plasma TMAO levels increased significantly, and were associated with cognitive and LTP decline, decreased functional connectivity, reduced neuronal plasticity, and dendritic spine density, along with higher levels of pro-inflammatory cytokines of IL-1β, IL-6, and TNF-α in the hippocampus after ischemia [49]. Administration of baicalin (50 and 100 mg/kg), a flavonoid from *Scutellaria baicalensis* with neuroprotective properties, could restore normal plasma levels of TMAO by modifying the composition of the intestinal microbiota, subsequently improve cognition and attenuate the neuropathology related to cerebral ischemia-reperfusion injury [49] (Table 3).

**Table 3 brainsci-12-01203-t003:** Main findings of TMAO in cerebral and cardiovascular disease.

Authors (year)	Subjects/Models	Interventions	Main Related Findings	Conclusions
Liu et al. (2020) [49]	2–3-month-old male C57BL/6J mice (*n* = 135)	Sham surgery, repeated global cerebral ischemia with the bilateral common carotid arteries 2 times, 3 times, and 4 times; intragastric administration of baicalin (25 mg/kg, 50 mg/kg, or 100 mg/kg); 200 μL solution of 0.5 g/L vancomycin, 1 g/L neomycin sulfate, 1 g/L metronidazole, 1 g/L ampicillin by gavage for 7 days	Plasma TMAO: increased by 3- and 4-repeated cerebral ischemia reperfusion, decreased by baicalin, correlated with behavioral and electrophysiological deficits Cognition (novel object recognition and Morris water maze): cognition improved by baicalin can be diminished with broad spectrum antibiotics Hippocampus: the reduction in Nissl bodies, dendritic spine density, and synaptic proteins, the higher levels of IL-1β, IL-6, and TNF-α after repeated cerebral ischemia-reperfusion can be restored by baicalin	Repeated cerebral ischemia-reperfusion would change gut microbiota composition, increase TMAO, reduce hippocampal neuronal plasticity, aggravate neuroinflammation, and cause cognitive decline that can be protected by oral supplementation with 50–100 mg/kg baicalin for 7 days after ischemia-reperfusion injury
Zhu et al. (2020) [44]	First-ever ischemic stroke patients admitted within 7 days of symptom onset (*n* = 256) Age- and sex-matched healthy controls (*n* = 100)	N/A	Plasma TMAO: obtained within 24 h after admission, detected by stable isotope dilution high-performance liquid chromatography with online tandem mass spectrometry, higher levels in stroke patients compared with healthy controls (5.6 ± 2.4 vs. 4.9 ± 1.8 μM, *p* = 0.012) Cognition: by MMSE at 1 year after stroke, MMSE score lowered with increasing quartile of TMAO level, the highest quartile of TMAO level was identified as an independent predictor for MMSE ≤ 26 after adjusting for potential confounders	Acute ischemic stroke patients who had higher TMAO levels would be at a higher likelihood of developing cognitive impairment 1 year after stroke
He et al. (2020) [50]	Older adults aged ≥65 years with cardiovascular disease	N/A	Plasma TMAO: higher in patients with cognitive frailty than those without (4.56 (2.81–7.59) vs. 3.38 (2.26–5.38) μM; *p* = 0.004), and each 2-unit increase in TMAO was associated with cognitive frailty after covariate adjustment	Elevated circulating TMAO levels were independently associated with physical and cognitive frailty among older adults with cardiovascular disease
Gong et al. (2021) [45]	Ischemic stroke patients admitted within 2 weeks of symptom onset with NIHSS score under 5 (*n* = 66 for TMAO analysis)	N/A	Plasma TMAO: obtained on the second day after admission, higher in patients with cognitive impairment (MoCA < 22) vs. those without cognitive dysfunction (MoCA ≥ 22) after 6 months of minor stroke onset (median 4.56 vs. 3.22 μM; *p* ≤ 0.001), while TMAO precursors of choline and L-carnitine levels had no differences	Higher plasma TMAO level at admission might suggest a potential marker of poststroke cognitive impairment
Zhong et al. (2021) [46]	Ischemic stroke patients admitted within 48 h of symptom onset with systolic blood pressure between 140 and 220 mmHg (*n* = 617)	N/A	Plasma TMAO: obtained within 24 h of admission, higher TMAO levels in participants who were older, had a higher prevalence of diabetes and a lower estimated glomerular filtration rate Plasma TMAO precursors (choline and betaine): inverse dose-response relationship with both MMSE and MoCA defined poststroke cognitive impairment Cognition: associated with plasma TMAO level when defined by MMSE < 27, but not MoCA < 25 (1 point was added for participants with education < 12 years)	Patients with higher plasma TMAO precursors of choline and betaine had lower risk of cognitive impairment 3 months after stroke, and the TMAO could only be considered as a risk factor when cognitive impairment was evaluated using the MMSE

Abbreviations: IL-1β, interleukin-1β; IL-6, interleukin-6; MMSE, mini-mental state examination; MoCA, Montreal cognition assessment; NIHSS, national institutes of health stroke scale; PD, Parkinson’s disease; TMAO, trimethylamine-N-oxide; TNF-α, tumor necrosis factor α.

## 5. Cognitive Frailty in Cardiovascular Disease

One study explored TMAO and cognitive frailty in older adults with cardiovascular disease [50]. In this cross-sectional setting, patients aged 65 years or older who had cardiovascular disease were assessed in terms of cognitive frailty, as determined by the simultaneous presence of physical frailty (three or more of the five conditions: weight loss, exhaustion, low activity, weakness, and slowness) and an MMSE score of ≤25; consequently, an independent association was observed between each 2-unit increase in TMAO and cognitive frailty (aOR, 1.21; 95% CI, 1.01–1.45; *p* = 0.04) [50] (Table 3).

## 6. Parkinson’s Disease Dementia

It is known that the pathology of Parkinson’s disease (PD) is linked to α-synuclein misfolding and aggregation, whereas TMAO has been shown to shift α-synuclein structures toward a more compact protein dimension in vitro [51]. As expected, PD patients with normal cognition had higher plasma TMAO levels than healthy controls (6.33 ± 0.56 vs. 3.76 ± 0.38 μM; *p* = 0.020) [52]. However, when comparing PD with normal cognition to patients with Parkinson’s disease dementia (PDD, diagnosed by an MMSE score of 25 or less combined with any impairment in the eight instrumental activities), no difference was detected in plasma TMAO levels (6.33 ± 0.56 vs. 10.61 ± 4.53 μM; *p* = 0.220) [52]. Neither was the TMAO cut-point of 4.88 μM (obtained using the Youden index) distinguishable for the longitudinal cognitive decline based on a sustained decrease of at least 2 points in the MMSE score (*p* = 0.191). Yet there was a multivariable-adjusted trend that higher baseline TMAO levels predicted the risk of cognitive deterioration after a mean period of 4.3 ± 2.2 years (adjusted hazard ratio, aHR, 1.001; 95% CI, 1.000–1.002; *p* = 0.091) [52] (Table 4).

However, it is still too early to determine if TMAO is a risk factor for PDD. There were conflicting findings on the association between TMAO and cognitive progression in PD patients in another longitudinal study with a mean follow-up duration of 2.75 ± 0.60 years [53]. In this study, PDD was diagnosed under the clinical diagnostic criteria, and briefly, cognitive impairment in at least two cognitive domains with abnormality in activities of daily living was required. In patients with drug-naïve, early-stage PD, the plasma TMAO level was not associated with the risk for PDD conversion (*p* = 0.488), but a lower baseline plasma TMAO level (<6.92 μM) tended to be independently associated with a lower risk for dementia conversion (aHR, 7.565; 95% CI, 1.004–57.019; *p* = 0.050). After removing cases with plasma TMAO levels that could be considered outliers, the conclusion was the same that higher baseline plasma TMAO levels tended to be associated with a lower risk for PDD conversion (aHR, 0.182; 95% CI, 0.028−1.174; *p* = 0.073) [53] (Table 4).

## 7. Postoperative Cognitive Dysfunction

In aged rats following major abdominal surgery [54] or anesthetic sevoflurane exposure [26], the elevated perioperative TMAO downregulated hippocampal antioxidant enzyme of methionine sulfoxide reductase A, increased the susceptibility to surgery-induced oxidative stress, and enhanced microglia-mediated neuroinflammation, contributing to exaggerations of cognitive deficit in the hippocampal-dependent memory consolidation (Table 5).

## 8. Current Gaps

Recent findings revealing the impact of aging on TMAO and the association between TMAO and the development and progression of NCD are intriguing (Table 1, Table 2, Table 3, Table 4 and Table 5), although many questions need to be solved. Most importantly, whether TMAO is a friend or foe of cognition is still uncertain. Observations are contradictory that cognition could be improved both after reduction of TMAO [27,36,37] and after supplementation of TMAO precursors [13,14,15,28]. Another concern arises from the evidence that TMAO has anti-inflammatory properties through beneficially regulating BBB integrity [24], but to date, most investigations have concluded the opposite; thus, the precise mechanisms of the cross-talk between TMAO and inflammation remain open. In clinical settings, it is consistent that TMAO increases with aging and in individuals with AD, but this feature might not be useful in the differential diagnosis or progression prediction for AD [29,33]. Additionally, it is yet to be determined whether TMAO affects cognitive aggravation in patients suffering from other types of NCD, such as PSCI and PDD. Further studies are required to explore if the baseline level of plasma TMAO is able to predict PSCI in stroke patients, and to rectify the conflicting effects of TMAO on cognitive decline in PD patients, better taking into account the sampling time of TMAO and the variations in defining criteria of cognitive progression.

## 9. Conclusions

Current evidence has demonstrated a link between TMAO and cognitive change due to aging and NCD. However, the nature of this link remains vague, because there is clear evidence not only for TMAO to influence the pathology underlying aging and NCD but also for these conditions to elevate the circulating TMAO level. We hypothesize that this truly is a bidirectional interaction; for example, aging or NCD itself may induce changes in plasma TMAO concentrations, and these changes may then increase the susceptibility to cognitive impairment. Studies to date have outlined potential mechanistic roles for such communication, often acting via neuroinflammation. However, it seems highly unlikely that TMAO is a driving force or the only player involved in determining variations in cognitive status in the context of aging and NCD. Looking forward, studies using preclinical models are required to assess the interaction between cognition and TMAO as well as its precursors, the producing enzyme FMO3, in different stages of NCD. Meanwhile, longitudinal clinical studies that dynamically assess the changes in TMAO levels with short-term and long-term cognitive trajectory would help to resolve uncertainty about the diagnostic potential of TMAO for cognitive aging and NCD. In short, the bulk of the evidence suggests that at least TMAO is increased with aging and AD, but more mechanistic understandings are needed to determine with certainty the effects of TMAO on aging and NCD-related cognitive impairment and progression.

## Figures and Tables

**Table 4 brainsci-12-01203-t004:** Main studies involving TMAO and Parkinson’s disease.

Authors (year)	Subjects/Models	Interventions	Main Related Findings	Conclusions
Chen et al. (2020) [52]	PD patients (*n* = 60) and healthy controls (*n* = 30)	N/A	Plasma TMAO: higher in PD patients than in controls, not differ between PD with normal cognition and PD with dementia Cognition: by MMSE, higher baseline TMAO levels associated with a trend for cognition deterioration during 4.3 ± 2.2 years	Plasma TMAO levels were elevated in patients with PD and correlated with disease severity and a trend of cognitive progression
Chung et al. (2020) [53]	Patients with drug-naïve early stage PD (*n* = 80) and healthy controls with normal cognition and without subjective cognitive impairment or a history of neurologic disease (*n* = 20)	N/A	Plasma TMAO: lower in PD patients than in controls, lower TMAO required higher doses of longitudinal dopaminergic medication for effective symptom control Cognition: PD dementia, diagnosed according to clinical diagnostic criteria, higher conversion risk predicted by lower plasma TMAO levels < 6.92 mmol/L with a borderline statistical significance (hazard ratio, 7.565; 95% confidence interval, 1.004–57.019; *p* = 0.050)	Baseline plasma TMAO levels have prognostic implications in patients with early-stage PD and that increasing TMAO levels may be helpful in slowing PD pathogenesis and PD dementia conversion

Abbreviations: PD, Parkinson’s disease; MMSE, mini-mental state examination; TMAO, trimethylamine-N-oxide.

**Table 5 brainsci-12-01203-t005:** Main studies of TMAO involved in postoperative cognitive dysfunction.

Authors (Year)	Subjects/Models	Interventions	Main Related Findings	Conclusions
Meng et al. (2019) [54]	Aged male F344xBN F1 rats (*n* = 14 each)	TMAO (120 mg/kg) in drinking water for 3 weeks vs. vehicle of tap water before surgery of laparotomy vs. sham operation	Plasma TMAO: increased before and 1 week after surgery Cognition (fear conditioning test): preexisting higher circulating TMAO reduced the freezing to context after surgery Hippocampus: preexisting higher circulating TMAO with surgery led to activated microglia↑, IL-1β↑, TNF-α↑, phospho-NF-κB p65↑, IκB-α↓, hydrogen peroxide levels↑, and antioxidant enzyme MsrA expression↓	The preexisting higher plasma TMAO led to cognitive impairment in aged rats after surgery or anesthetic sevoflurane exposure by microglia-mediated neuroinflammation, enhanced oxidative stress, and decreased MsrA expression
Zhao et al. (2019) [26]	20-month-old male SD mice	120 mg/kg TMAO in drinking water vs. tap water for 3 weeks, exposed to 2.6% sevoflurane in 30% oxygen or 30% oxygen only in an anesthetic induction chamber for 4 h after 2 weeks	Plasma TMAO: increased after 2 weeks of TMAO treatment Cognition (fear conditioning test): preexisting higher circulating TMAO reduced the freezing to context after sevoflurane exposure Hippocampus: preexisting higher circulating TMAO with sevoflurane exposure led to activated microglia↑, IL-1β↑, TNF-α↑, NADPH oxidase activity↑, H_2_O_2_ levels↑, and antioxidant enzyme MsrA expression↓	

Abbreviations: ↑, increase; ↓, decrease; IκB-α, inhibitor of nuclear factor-κB; IL-1β, interleukin-1β; MsrA, methionine sulfoxide reductase A; NADPH, nicotinamide adenine dinucleotide phosphate; phospho-NF-κB, phospho-nuclear factor-κB; TMAO, trimethylamine-N-oxide; TNF-α, tumor necrosis factor α.

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
