# Peer review of "Association of Trimethylamine N-Oxide with Normal Aging and Neurocognitive Disorders: A Narrative Review"

_brainsci, 2022, doi:10.3390/brainsci12091203_

Round 1

Reviewer 1 Report

This short review article summarizes the TMAO’s cognitive association with aging, Alzheimer’s disease, stroke, and cardiovascular disease and points out the lack of mechanistic understanding.

Here are my minor comments.

Page 2 Paragraph 4 Line 3: TMAO at what level induces brain aging-related cognitive dysfunction in mice?

Page 2 Paragraph 4 Line 3: synaptic plasticity-related proteins abbreviate SYN, PSD-95, and NMDAR1.

Page 2 Paragraph 5 Line 12: Abbreviate MMSE as first introduced the term, but it is abbreviated later in Page 8 Paragraph 1 Line 4

Page 8 Paragraph 3 Line 6: what is baicalin? Why it is administered. Little detail is needed.

Author Response

We appreciate the reviewer for taking the time to review the manuscript and give helpful suggestions and comments. We have carefully studied the reviewer’s comments, made corrections accordingly, and improved the English writing in the revised version. Please find below a point-by-point response to the reviewer’s points.

Point 1: Page 2 Paragraph 4 Line 3: TMAO at what level induces brain aging-related cognitive dysfunction in mice?

Response 1: Thank you for the thoughtful comment. Compared with the control groups, mice in the TMAO treatment groups were given TMAO (Sigma, USA) dissolved in water at a concentration of 1.5% for 16 weeks, and they could lead to aggravated brain aging and cognitive function. In the revised manuscript, we have modified this sentence as “In mice, 16-week treatment of TMAO at a concentration of 1.5% could induce and aggravate brain aging and aging-related cognitive dysfunction.”

Point 2: Page 2 Paragraph 4 Line 3: synaptic plasticity-related proteins abbreviate SYN, PSD-95, and NMDAR1.

Response 2: We apologize for the misunderstanding of the abbreviation. The synaptic plasticity-related proteins, including synaptophysin (SYN), postsynaptic density-95 (PSD95), and N-methyl-D-aspartate receptor subunit 1 (NMDAR1) were measured. In this revised version, the “SYN, PSD-95, and NMDAR1” following “synaptic plasticity-related proteins” in the brackets were deleted to avoid confusion.

Point 3: Page 2 Paragraph 5 Line 12: Abbreviate MMSE as first introduced the term, but it is abbreviated later in Page 8 Paragraph 1 Line 4

Response 3: Thanks for pointing this out. In the revised version, MMSE has been introduced at first appearance, and the abbreviation has been used in the rest of the manuscript.

Point 4: Page 8 Paragraph 3 Line 6: what is baicalin? Why it is administered. Little detail is needed.

Response 4: Thank you for this suggestion. Baicalin is one of the active flavonoids in the roots of Scutellaria baicalensis with anti-tumor, anti-viral, anti-microbial, anti-inflammatory, antioxidative, and neuroprotective properties. Thus, we have added, “a flavonoid from Scutellaria baicalensis with neuroprotective properties” to explain baicalin in this revised manuscript.

Reviewer 2 Report

The review entitled “Association of Trimethylamine N-oxide with Normal Aging and Neurocognitive Disorders: A Narrative Review”, provides in relation to cognitive change due to aging and NCD. It is well written and detailed.

Author Response

We gratefully thank you for the precious time spent making constructive remarks on our manuscript. Considering the suggestion on English writing and style, we have carefully checked and improved the English writing, and thoroughly polished the manuscript. We hope this revised version will meet the standard of this journal.

Reviewer 3 Report

Reviewer’s comments:

The manuscript by Junshan Zhou and co-workers is dedicated to an important issue for the neuroscience: whether or not trimethylamine-N-oxide (TMAO) could be used as biomarker of cognitive dysfunction. The presented review article reveals an important issue, since the increasing agе of the population is a high risk factor for neurocognitive disorders such as Alzheimer’s disease, Parkinson’s disease, dementia, post-stroke cognitive impairment. The whole study is valuable, literature sources are summarized and analyzed correctly, the author’s point of view is presented in a clear manner. In general, the manuscript contains interesting material.

Author Response

Thank you very much for your positive comments on our work. We appreciate your dedicated time and effort in providing feedback on our manuscript. We have carefully checked and improved the English writing in the revised version. Please see if the revised paper meets the English presentation standard.